# Regulation of Light Absorption and Energy Dissipation in Sweet Sorghum Under Climate-Relevant CO_2_ and Temperature Conditions

**DOI:** 10.3390/biology14091185

**Published:** 2025-09-03

**Authors:** Jin-Jing Li, Li-Hua Liu, Zi-Piao Ye, Chao-Wei Zhang, Xiao-Long Yang

**Affiliations:** 1Institute of Biotechnodogy, Gansu Academy of Agricultural Sciences, Lanzhou 730070, China; ljj0623@gsagr.cn; 2College of Safety Engineering and Emergency Management, Nantong Institute of Technology, Nantong 226002, China; liulihua032007@126.com; 3Math & Physics College, Jinggangshan University, Ji’an 343009, China; yezp@jgsu.edu.cn; 4School of Life Sciences, Nantong University, Nantong 226019, China; 5State Key Laboratory of Environmental Chemistry and Ecotoxicology, Research Center for Eco-Environmental Sciences, Chinese Academy of Sciences, Beijing 100085, China

**Keywords:** electron transport rate, quantum efficiency of photosystem II, light response, photosynthetic pigments, CO_2_ concentration

## Abstract

Climate change is driving increases in both atmospheric CO_2_ and temperature, posing challenges for plant growth and productivity. In this study, we examined how sweet sorghum, an important bioenergy crop, optimizes sunlight utilization under these future climatic conditions. Plants were cultivated under different combinations of CO_2_ concentration and temperature, and their light absorption and energy-use patterns were evaluated. We found that elevated CO_2_ enhanced light capture and improved its use for photosynthetic growth, whereas higher temperatures activated the plant’s protective mechanisms, enabling the safe dissipation of excess energy as heat. These coordinated strategies illustrate the inherent resilience of sweet sorghum. Our findings provide insights for the development of crop varieties capable of sustaining high yields and contributing to food and bioenergy security under climate change.

## 1. Introduction

Photosynthesis is a fundamental physiological process essential for biomass production in plants. It has been widely reported that 90–95% of crop yield derives from photosynthesis, with the remaining 5–10% relying on nutrient uptake through the root system [1,2]. The process begins with the absorption of light energy by light-harvesting pigment–protein complexes located on the thylakoid membranes within chloroplasts. When photons are absorbed, antenna pigments transition from the ground state to an excited state, an unstable condition that requires rapid energy dissipation [3]. Under normal physiological conditions, most of this excitation energy is transferred among chlorophyll molecules via resonance and ultimately reaches the reaction centers *P*_680_ (photosystem II, PSII) and *P*_700_ (photosystem I, PSI) [4,5]. Excited *P*_680_ initiates water splitting, leading to charge separation and triggering the electron transport chain to generate NADPH, together with ATP synthesis driven by the proton gradient. These energy carriers fuel CO_2_ fixation in the Calvin cycle.

However, not all absorbed light energy is utilized for photochemical reactions. Excited pigment molecules may also dissipate excess energy as heat or emit chlorophyll fluorescence [4,6]. The partitioning of excitation energy among photochemistry, fluorescence, and non-photochemical quenching (NPQ) is dynamically regulated by environmental conditions such as light intensity [7], temperature, water availability, CO_2_ concentration [8], and nutrient status [9]. This flexible allocation is a key adaptive mechanism that allows plants to sustain efficient photosynthesis and avoid photodamage [7,10]. Light-harvesting pigment molecules are central to this regulation, functioning as the primary sensors of light stress [11]. Alterations in their functional state serve as early signals for photoprotective activation, ultimately determining the upper limits of light-use efficiency [5,12]. A detailed understanding of the absorption properties of these pigments is therefore essential for clarifying the dynamic regulation of excitation energy distribution, and has significant implications for improving photosynthetic efficiency and crop yield potential.

Sweet sorghum (*Sorghum bicolor* (L.) Moench), a C_4_ graminaceous crop, is valued for its strong environmental adaptability and high stem sugar content, making it an important resource for bioenergy, animal feed, sugar production, and the food industry [13,14]. Its deep root system and tolerance to drought and salinity allow it to grow on marginal lands across Asia, Africa, the Americas, and Europe. Major producers include China, India, the United States, Brazil, and several African nations. Sweet sorghum typically grows to 1.2–4 m in height and achieves stem yields of 47.4–52.1 t ha^−1^ year^−1^, juice extraction rates of 59–65.4%, juice sugar content of 16.1–19.5 °Brix, and grain yields of 1.8–5.0 t ha^−1^ [15]. According to FAO statistics, global sorghum production reached 57 million tons in 2023 [16], demonstrating its bioeconomic potential. However, the escalating impacts of climate change, particularly the rise in atmospheric CO_2_ concentrations and global temperatures, pose significant challenges to the photosynthetic capacity and growth dynamics of sweet sorghum. Despite its inherent C_4_ advantages under high light and temperature, prolonged exposure to combined CO_2_ enrichment and heat stress may disrupt photosynthetic enzyme systems, impair stomatal regulation, and alter carbon–nitrogen metabolism. These disruptions can ultimately affect growth duration, sugar accumulation, and yield stability.

Building upon our previous research, which characterized the leaf-scale gas exchange responses (net photosynthetic rate *A*_n_, stomatal conductance *g*_S_, transpiration rate *T*_r_, and water-use efficiency WUE) of sweet sorghum to varying light, CO_2_, and temperature [13], this study delves deeper into the underlying photobiological mechanisms. Our earlier work established how these environmental factors affect the overall carbon and water flux, but did not address the molecular-scale processes governing light energy absorption, transfer, and dissipation. Plants exposed to elevated CO_2_ and high-temperature conditions often experience reduced photochemical efficiency [8]. Although increased CO_2_ may enhance carbon assimilation, excessive heat stress destabilizes thylakoid membranes and disrupts electron transport [17]. This, combined with inhibited RuBisCO activity and reduced regeneration of ribulose-1,5-bisphosphate, decreases the consumption of energy equivalents (ATP and NADPH) within CO_2_ fixation [17,18]. As a result, plants must dissipate excess excitation energy and rely on alternative electron sinks to maintain redox balance and photoprotection. Light-harvesting pigment molecules respond dynamically to these abiotic stresses by modulating energy capture and allocation. For example, elevated CO_2_ can reduce chlorophyll content per unit leaf area in *Oryza sativa* L. [19] and decrease the intrinsic light absorption cross-section (*σ*_ik_) in *Glycine max* L. (Merr.) [20], thereby lowering the risk of over-excitation under strong light. Similarly, high temperatures accelerate reorganization of light-harvesting complexes, enhancing non-photochemical quenching (NPQ) via activation of the xanthophyll cycle and PsbS protein aggregation [12]. These adjustments minimize reactive oxygen species by channeling excess energy into heat dissipation while maintaining basal electron transport for carbon assimilation [18]. The capacity to dynamically adjust *σ*_ik_, NPQ, and excitation lifetime is thus essential for maintaining stable photosynthesis under fluctuating environments. Yet, quantitative characterization of these mechanisms in C_4_ species remains limited.

Chlorophyll fluorescence techniques provide a non-invasive, sensitive, and real-time method for assessing the photosynthetic status of plants under environmental stress [4,6]. In this study, a LI-6800 portable photosynthesis system equipped with a fluorescence chamber was used to measure light response curves of the electron transport rate (*J*), the effective quantum yield of PSII (Φ_PSII_), and NPQ in sweet sorghum leaves under various combinations of CO_2_ concentration and temperature. By integrating these datasets with leaf chlorophyll content and applying photosynthetic mechanistic models developed by Ye et al. [21], we quantified key molecular-scale parameters, including the total number of antenna pigment molecules in the measured leaf (*N*_0_), their eigen-absorption cross-section (*σ*_ik_), the minimum average excited-state lifetime (*τ*_min_), effective light energy absorption cross-section (*σ′*_ik_), and the number of excited-state pigment molecules (*N*_k_). The aim of this study was to clarify how environmental factors regulate excitation energy capture and transfer in sweet sorghum, and to reveal its light-response dynamics under climate-relevant CO_2_ and heat stress conditions.

## 2. Materials and Methods

### 2.1. Plant Material

Seedlings of sweet sorghum (*Sorghum bicolor* L. Moench, KFJT-1) were propagated following methods established in earlier work [13]. After germination, the seedlings were relocated to plastic containers and cultivated in a growth chamber (RDN-1000E-4, Ningbo Dongnan Instrument Co., Ltd., Ningbo, China) under controlled environmental conditions. These conditions included a light intensity of 350 μmol photons m^−2^ s^−1^, a consistent temperature of 25 °C, and a photoperiod of 16 h of light followed by 8 h of darkness. When seedlings developed eight fully expanded leaves, uniformly growing, healthy plants were selected for subsequent measurements.

### 2.2. Light Response Measurement

Chlorophyll fluorescence was measured using an LI-6800 portable photosynthesis system (Li-Cor Inc., Lincoln, NE, USA) equipped with a LI-6800-01A fluorometer chamber. Measurements were conducted on clear days between 8:30–11:30 AM and 2:00–5:30 PM using automated protocols. Leaves were carefully clamped in the chamber and exposed to 1800 μmol photons m^−2^ s^−1^ photon flux density (*I*) for 40 min to activate photosynthetic enzymes. The light response was then determined by stepwise decreases in irradiance, starting at 2000 μmol photons m^−2^ s^−1^ and progressing through 1800, 1600, 1400, 1200, 1000, 800, 600, 400, 200, 150, 100, 50, 25, and finally 0 μmol photons m^−2^ s^−1^. At each irradiance level, data were recorded after 2–3 min of stabilization. To ensure accuracy, the system automatically calibrated the reference and sample chambers before recording. Measurements were carried out under two leaf temperatures (30 °C and 35 °C) to simulate normal and heat-stress conditions associated with climate change. CO_2_ concentrations in the chamber were set to 250, 410, and 550 μmol mol^−1^, supplied from an external gas cylinder through the injection module. The mixed-gas flow rate was regulated at 500 μmol s^−1^ by the LI-6800 system. Each measurement was performed in triplicate to minimize variability and ensure data reliability. After completion, the same leaves were collected for chlorophyll quantification.

### 2.3. Chlorophyll Quantification

For pigment analysis, a 1 cm^2^ section from the previously measured leaf area was excised, weighed, and chopped into approximately 1 mm^2^ fragments. Samples were immersed in 5 mL of 80% acetone within sealed test tubes and kept in darkness at low temperature for 24 h to ensure complete extraction, aided by gentle intermittent shaking. After decolorization, the mixture was centrifuged at 4000 rpm for 5 min to obtain a clear supernatant. Chlorophyll concentration per leaf area unit (mg m^−2^) was calculated from absorbance readings at 663 nm and 645 nm according to Wellburn’s method [22].

### 2.4. Model Fitting and Computational Analysis

#### 2.4.1. *J*–*I* Mechanistic Model

Ye et al. [21] proposed a mechanistic framework for modeling the *J*–*I* relationship, providing a biophysically grounded alternative to empirical models such as the negative exponential and non-rectangular hyperbolic functions [23].

This model describes the sequence of events in photosynthesis—from light capture through energy transfer—based on the properties of pigment–protein complexes. The governing equation is:(1)$$J=\alpha_{e}\frac{1-\beta_{e}I}{1+\gamma_{e}I}I\;$$
where *α*_e_ denotes the initial slope of the *J*–*I* curve, while *β*_e_ and *γ*_e_ are parameters accounting for biophysical interactions, including the degeneration of energy levels in photosynthetic pigments (ground and excited states), the occupation probabilities of photochemistry, heat loss, and fluorescence emission, and the rates of photochemical efficiency and heat dissipation. The model is implemented in the *Photosynthesis Model Simulation Software* (PMSS, Jinggangshan University, Ji’an, China) (http://www.zipiao.tech/, in Chinese/English version, accessed on 11 August 2024) to fit experimental *J*–*I* data and extract parameter estimates.

By integrating leaf chlorophyll content into the *J*–*I* model, we can estimate additional pigment-related traits, including the total number of pigment molecules per leaf (*N*_0_), their eigen-absorption cross-section (*σ*_ik_), and the minimum average excited-state lifetime (*τ*_min_) [24].

The saturation light intensity (*I*_sat_) is derived by setting the first derivative of Equation (1) to zero:(2)$$I_{sat}=\frac{\sqrt{\frac{{(\beta }_{e}+\gamma_{e})}{\beta_{e}}}-1}{\gamma_{e}}\;$$

Subsequently, the maximum electron transport rate (*J*_max_) can be calculated as:(3)$$J_{max}=\alpha_{e}\left({\left.\frac{\sqrt{\beta_{e}+\gamma_{e}}-\sqrt{\beta_{e}}}{\gamma_{e}}\right)}^{2}\right.\;$$

#### 2.4.2. Mechanistic Models for Φ_PSII_, σ′_ik_, and N_k_

Φ_PSII_ reflects the efficiency with which absorbed photons are converted into chemical energy. Building on the link between *J* and Φ_PSII_, Yang et al. [23] developed mechanistic models for Φ_PSII_, the effective light energy absorption cross-section of antenna pigment molecules (*σ′*_ik_), and the number of excited-state pigment molecules (*N*_k_) as functions of *I*. These relationships are defined as follows:(4)$$\Phi_{PSII}=\Phi_{PSIImax}\frac{1-\beta_{e1}I}{1+\gamma_{e1}I}\;$$
where $\Phi_{PSIImax}=\frac{\alpha_{e}}{\alpha \beta }$, with *α* = 0.5 representing the partitioning of absorbed light between PSII and PSI [25], and *β* = 0.84 the assumed leaf absorptance [26]. When *I* = 0 μmol photons m^−2^ s^−1^, *β*_e1_ and *γ*_e1_ equal *β*_e_ and *γ*_e_ in Equation (1), and Φ_PSII_ = Φ_PSIImax_. However, their values differ when *I* > 0. Equation (4) incorporates the quantitative relationship between Φ_PSII_ and the intrinsic characteristics of light-harvesting pigment molecules [23].

Light absorption efficiency per unit pigment, *σ′*_ik_, varies with light intensity as:(5)$$\sigma_{\mathrm{ik}}^{\prime }=\frac{1-\beta_{e}I}{1+\gamma_{e}I}\sigma_{\mathrm{ik}}\;$$

Combining Equations (4) and (5) gives:(6)$$\Phi_{PSII}=\Phi_{PSIImax}\frac{\sigma_{\mathrm{ik}}^{\prime }}{\sigma_{\mathrm{ik}}}\;$$

Under stable environmental conditions and fixed species traits, Φ_PSII_ shows a direct proportionality to *σ′*_ik_.

Meanwhile, Ye [24] proposed that, at any given *I*, the total pigment pool (*N*_0_) is partitioned into excited-state (*N*_k_) and ground-state pigments (*N*_i_):(7)$$N_{k}=\;\frac{1}{1-\frac{g_{i}}{g_{k}}}\frac{\beta_{e}I}{1+\gamma_{e}I}N_{0}\;$$
and(8)$$N_{i}=(1-\frac{1}{1-\frac{g_{i}}{g_{k}}}\frac{\beta_{e}I}{1+\gamma_{e}I})N_{0}\;$$

These equations illustrate that both *N*_k_ and *N*_i_ dynamically respond to light intensity. As *I* increases, *N*_k_ rises while *N*_i_ decreases, indicating a reciprocal relationship. In complete darkness (*I* = 0), all pigment molecules remain in the ground state (*N*_k_= 0, *N*_i_ = *N*_0_).

### 2.5. Statistical Analysis

All parameters are reported as means ± *SE* from three biological replicates. Two-way ANOVA was used to test treatment effects, and paired-sample *t*-tests (*p* < 0.05) were applied to compare model outputs with observed data. Model performance was evaluated using the coefficient of determination (*R*^2^) obtained from SPSS version 24.0 (SPSS Inc., Chicago, IL, USA).

## 3. Results

### 3.1. J–I Response Curves Under Different CO_2_ and Temperature Conditions

The electron transport rate (*J*) in sweet sorghum leaves exhibited typical light-response patterns under all six combinations of ambient CO_2_ and temperature (Figure 1). *J* increased rapidly with rising *I* before reaching a plateau at high light. At 30 °C (Figure 1A,B), the *J–I* curves under low (250 μmol mol^−1^) and ambient CO_2_ (410 μmol mol^−1^) were similar, with no significant increase in *J*. In contrast, elevated CO_2_ (550 μmol mol^−1^) significantly increased *J* across the irradiance gradient (Figure 1C), resulting in higher *J*_max_ and *I*_e-sat_ values (Table 1). At 35 °C (Figure 1D–F), *J* increased progressively with CO_2_ enrichment, and the *J–I* curves shifted upward. Under low CO_2_ (250 μmol mol^−1^), a slight decline in *J* was observed beyond light saturation at both temperatures (Figure 1A,D), indicating PSII dynamic down-regulation. However, this decline was absent under elevated CO_2_ (Figure 1C,F), suggesting that high CO_2_ mitigates photoinhibition and improves electron transport capacity under heat stress. Model-simulated *J*_max_ and *I*_e-sat_ closely matched measured data (*R*^2^ > 0.997), with no significant differences between fitted and observed values (*p* > 0.05).

Table 1 quantitatively confirms these trends. At 30 °C, *J*_max_ rose from 133.63 ± 3.30 μmol electrons m^−2^ s^−1^ (250 μmol mol^−1^ CO_2_) to 184.82 ± 5.49 μmol electrons m^−2^ s^−1^ (550 μmol mol^−1^ CO_2_). Similarly, *I*_e-sat_ increased from 1280.03 ± 48.99 μmol photons m^−2^ s^−1^ to 1800.00 ± 0.00 μmol photons m^−2^ s^−1^, demonstrating enhanced light utilization under elevated CO_2_. At 35 °C, *J*_max_ increased from 142.20 ± 1.92 mol electrons m^−2^ s^−1^ to 223.57 ± 5.06 μmol electrons m^−2^ s^−1^ across the same CO_2_ gradient, while *I*_e-sat_ rose from 1400.02 ± 63.24 μmol photons m^−2^ s^−1^ to 2000.00 ± 0.00 μmol photons m^−2^ s^−1^. Integration of chlorophyll content data (425.90 ± 1.67 mg m^−2^) into the *J*–*I* model revealed a decline in the eigen-absorption cross-section (*σ*_ik_) with rising CO_2_, especially at 35 °C (from 2.58 ± 0.05 × 10^−21^ m^2^ to 2.16 ± 0.09 × 10^−21^ m^2^). This suggests down-regulation of pigment light absorption to prevent over-excitation. The minimum excited-state lifetime (*τ*_min_) also decreased significantly, implying faster energy dissipation and turnover. In line with these changes, the number of pigment molecules per unit area (*N*_0_) decreased with elevated CO_2_, particularly under heat stress.

### 3.2. Φ_PSII_–I Response Curves Under Different CO_2_ and Temperature Conditions

As shown in Figure 2, Φ_PSII_ decreased non-linearly with increasing *I*. The mechanistic model reproduced the observed PSII–I responses well, although some deviations occurred at maximum Φ_PSII_ (Φ_PSIImax_) (Table 1). Temperature exerted minimal influence on the light response of Φ_PSII_. At a fixed CO_2_, Φ_PSII_–*I* curves at 30 °C and 35 °C were nearly identical, with similar Φ_PSIImax_ values (Table 1). In contrast, CO_2_ enrichment strongly enhanced Φ_PSII_ at all light levels. At low CO_2_ (250 μmol mol^−1^), Φ_PSII_ dropped rapidly as *I* increased (Figure 2A,D). With increasing CO_2_, this decline slowed markedly (Figure 2B,E). At 550 μmol mol^−1^ CO_2_, Φ_PSII_ remained above 0.2 even at 2000 μmol photons m^−2^ s^−1^ (Figure 2C,F), indicating sustained PSII functionality under high light and elevated CO_2_.

### 3.3. σ′_ik_–I Response Curves Under Different CO_2_ and Temperature Conditions

Antenna pigment molecules absorb incoming light and transition into higher energy states. The effective absorption cross-section of antenna pigments (*σ′*_ik_) quantifies their photon-capturing efficiency under varying irradiances. As shown in Figure 3, *σ′*_ik_ in sweet sorghum decreased steadily with increasing *I*.

The *σ′*_ik_–*I* curves exhibited similar trends to the Φ_PSII_–*I* curves (Figure 2A–F), indicating a strong correlation between light absorption and photosynthetic efficiency. Elevated temperatures significantly increased *σ′*_ik_, especially under ambient (Figure 3E) and high (Figure 3F) CO_2_ conditions. For example, at 1800 μmol photons m^−2^ s^−1^, the temperature-induced increase in *σ′*_ik_ was only 4.5% at 250 μmol mol^−1^ CO_2_, but reached 33.8% and 21.1% at 410 μmol mol^−1^ and 550 μmol mol^−1^ CO_2_, respectively. CO_2_ enrichment also altered the response pattern of *σ′*_ik_ to *I*. At 250 and 410 μmol mol^−1^ CO_2_, *σ′*_ik_ decreased non-linearly with *I* (Figure 3A,B,D,E). In contrast, at 550 μmol mol^−1^, *σ′*_ik_ showed approximately linear decline (Figure 3C,F). Overall, raising CO_2_ from 250 to 550 μmol mol^−1^ increased *σ′*_ik_ by 41.8% at 30 °C and 64.3% at 35 °C under high irradiance of 1800 μmol photons m^−2^ s^−1^.

### 3.4. NPQ–I and N_k_–I Response Curves Under Different CO_2_ and Temperature Conditions

Non-photochemical quenching (NPQ) functions as a key photoprotective mechanism that prevents photooxidative damage by dissipating excess excitation energy as heat. As shown in Figure 4, NPQ increased non-linearly with irradiance and approached saturation under high light across all treatments.

Elevated temperatures significantly enhanced NPQ. CO_2_ concentration exerted a dual effect. NPQ decreased when CO_2_ increased from 250 to 410 μmol mol^−1^, but rose again as CO_2_ was further elevated to 550 μmol mol^−1^. For instance, at 1800 μmol m^−2^ s^−1^, increasing CO_2_ from 250 to 410 μmol mol^−1^ reduced NPQ by 0.05 (30 °C) and 0.06 (35 °C) (Figure 4A,B,D,E). Conversely, elevating CO_2_ from 410 to 550 μmol mol^−1^ increased NPQ by 0.06 (30 °C) and 0.19 (35 °C) (Figure 4B,C,E,F).

NPQ is closely linked to the de-excitation of pigment molecules in the excited state (*N*_k_). Figure 5 presents the *N*_k_–*I* curves. Under low CO_2_ (250 μmol mol^−1^), *N*_k_ increased non-linearly with *I* at two temperatures (Figure 5D). Raising CO_2_ from 250 to 410 μmol mol^−1^ had little effect at 30 °C (Figure 5A,B), but at 35 °C, *N*_k_ decreased significantly, showing an almost linear increase with *I* (Figure 5D,E). This trend became more pronounced at 550 μmol mol^−1^ CO_2_ (Figure 5C,F). Overall, both elevated temperature and CO_2_ significantly reduced *N*_k_. At 1800 μmol photons m^−2^ s^−1^, increasing CO_2_ from 410 to 550 μmol mol^−1^ lowered *N*_k_ by about 33% at 30 °C and ~20% at 35 °C, indicating enhanced dissipation of excitation energy under these conditions.

## 4. Discussion

This study extends our previous work on sweet sorghum, which characterized leaf-scale gas exchange responses (e.g., *A*_n_, *g*_s_, and WUE) to light, CO_2_, and temperature [13]. Here, by integrating chlorophyll fluorescence with a mechanistic biophysical model, we elucidated the molecular-scale mechanisms underlying those leaf-level responses. Our findings reveal how sweet sorghum coordinates CO_2_ availability, temperature, and irradiance to sustain photosynthetic function, thereby providing a mechanistic explanation for previously observed physiological patterns.

### 4.1. Regulation of Electron Transport and Light Utilization

Previous studies have shown that elevated CO_2_ enhances photosynthetic rates in C_4_ plants by improving phosphoenolpyruvate carboxylase (PEPcase) efficiency and reducing photorespiratory losses [8,27,28]. Sales et al. [27], for instance, reported increased *J*_max_ in maize and sorghum under elevated CO_2_, though they did not quantify changes in light-saturation dynamics. Our results confirm and extend these findings. Elevated CO_2_ not only boosted *J*_max_ (by 57% at 35 °C) but also increased the saturation irradiance (*I*_e-sat_), thereby improving the capacity to exploit high irradiance efficiently. This led to greater consumption of ATP and NADPH in carbon assimilation, consistent with the observed increase in *J*_max_ (223.57 ± 5.06 μmol electrons m^−2^ s^−1^ at 35 °C/550 μmol mol^−1^ CO_2_ vs. 142.20 ± 1.92 μmol electrons m^−2^ s^−1^ at 35 °C/250 μmol mol^−1^ CO_2_).

Dynamic down-regulation of PSII is an adaptive reduction in photochemical efficiency triggered by supra-saturating irradiance, mediated largely through energy-dependent quenching (qE) and sustained NPQ [29,30]. This mechanism protects the electron transport chain by dissipating excess energy and maintaining redox balance [31]. In our study, photoinhibition and PSII down-regulation were pronounced under low CO_2_, but were strongly suppressed under elevated CO_2_. Although high temperature typically disrupts the electron transport chain and induces redox imbalance, particularly under combined high light and heat stress [18,32], our results indicate that elevated CO_2_ partially alleviates this inhibition at 35 °C. Specifically, the upward shift in the *J–I* curve and sustained *J* under high CO_2_ suggest improved plastoquinone (PQ) turnover and redox balance, delaying photoinhibition. The significant increase in *J*_max_ under elevated CO_2_, even at 35 °C, reflects enhanced consumption of ATP and NADPH by the Calvin cycle. This increased metabolic demand likely prevents over-reduction of electron carriers such as the PQ pool and ferredoxin by providing an efficient sink for photosynthetic products. In contrast, elevated temperature alone intensified excitation pressure, as evidenced by increased NPQ and decreased *N*_k_ in the present study. However, the synergistic effect between high CO_2_ and temperature appeared to mitigate this effect through stimulated carbon assimilation, which consumed more reductants and helped maintain linear electron flow. Thus, sweet sorghum appears to exploit elevated CO_2_ to stabilize electron transport and counteract heat-induced stress, reducing reliance on photoprotective energy dissipation.

### 4.2. Regulation of Excitation Energy Allocation and Photoprotection via Antenna Pigment Dynamics

Our findings demonstrate that elevated CO_2_ and temperature significantly alter antenna pigment behavior, leading to improved regulation of excitation energy. Specifically, the eigen-absorption cross-section (*σ*_ik_) and minimum excited-state lifetime (*τ*_min_) decreased under elevated CO_2_, suggesting faster excitation turnover and reduced risk of over-excitation—consistent with previous findings in *Glycine max* [20].

Importantly, the mechanistic model employed in this study allowed in vivo quantification of the effective light absorption cross-section (*σ′*_ik_), providing a direct link to the efficiency of PSII photochemistry (Φ_PSII_). Unlike empirical models [33,34,35,36,37], our approach reveals that Φ_PSII_ and *σ′*_ik_ exhibit strong co-regulation. CO_2_ enrichment significantly increased *σ′*_ik_—by up to 64% under high light and temperature—and maintained Φ_PSII_ above 0.2 even at 2000 µmol m^−2^ s^−1^. These results support the view that elevated CO_2_ enhances excitation energy capture per pigment molecule, potentially through improved antenna organization and reduced exciton loss [12,38,39].

Elevated temperature also amplified *σ′*_ik_, especially under ambient and high CO_2_ (e.g., a 33.8% rise at 410 μmol mol^−1^ CO_2_), suggesting thermally induced reorganization of antenna complexes. Declines in Φ_PSII_ with irradiance paralleled reductions in *σ′*_ik_, reflecting a dynamic trade-off between photochemistry, thermal dissipation, and fluorescence. Recent spectroscopic work by Leiger et al. [40] reinforces this interpretation, demonstrating that the red-tail absorption of chlorophyll *a*—linked to the *Q*_y_ transition—is thermally activated and diminishes under cooling, implying that pigment optical properties are highly sensitive to vibrational coupling and protein–environment interactions. This supports the idea that elevated temperature modulates *σ′*_ik_ through altered vibronic states, thereby enhancing energy dissipation capacity.

In addition, NPQ was enhanced by both rising temperature and CO_2_. For instance, at 35 °C, NPQ increased by 0.19 units when CO_2_ rose from 410 to 550 μmol mol^−1^, suggesting a CO_2_-enhanced non-radiative energy dissipation under heat stress. This pattern aligns with previous findings that thermal and pH shifts promote NPQ through activation of the xanthophyll cycle and conformational changes in LHCII–PsbS complexes [12,41]. In addition, our study contributes a novel mechanistic insight by quantifying *N*_k_, the number of pigment molecules in the excited state. *N*_k_ serves as a key indicator of excitation pressure, determining the allocation of energy between photochemical quenching and photoprotective dissipation pathways [21,42]. This often-overlooked parameter decreased significantly under elevated CO_2_ and temperature—by approximately 33% at 30 °C and 20% at 35 °C—establishing a strong inverse relationship with NPQ. These results support the hypothesis that the dissipation of excess excitation energy via NPQ originates from the deactivation of excited pigment states, thereby lowering ROS risk and safeguarding PSII under stress [41].

Together, these pigment-level adjustments highlight a finely tuned regulatory system in which sweet sorghum modulates excitation energy allocation in response to CO_2_ and thermal conditions, balancing carbon fixation with dynamic photoprotection. This finely tuned regulation at the molecular level contributes to the crop’s robust adaptation to climate-related stresses.

### 4.3. Implications for Climate Resilience in C_4_ Crops

Our findings underscore the plasticity of sweet sorghum’s photosynthetic machinery in adjusting to simultaneous CO_2_ and temperature stress. The combined enhancements in *J*_max_, *σ′*_ik_, and NPQ, along with reductions in *N*_k_, illustrate how the species dynamically balances energy use and dissipation. This plasticity has important breeding implications. Selection for genotypes that maintain high *σ′*_ik_ and stable Φ_PSII_ under heat and CO_2_ enrichment may improve resilience and yield. Moreover, the mechanistic parameters identified here (*σ′*_ik_, *N*_k_, *τ*_min_) can serve as physiological markers in high-throughput phenotyping pipelines. However, it is important to note that the responses reported in this study reflect short-term acclimation to sudden changes in CO_2_ and temperature, as the plants were not pre-adapted to these stress conditions. While these immediate mechanistic responses are highly informative for understanding the initial biophysical adjustments of photosynthesis, they may differ from long-term adaptations that involve changes in gene expression, leaf morphology, and nutrient allocation. For instance, long-term exposure to elevated CO_2_ can lead to photosynthetic acclimation via carbohydrate accumulation and nitrogen dilution [14]. Therefore, field studies employing Free-Air CO_2_ Enrichment (FACE) or open-top chambers are essential to validate these mechanisms under realistic conditions where plants undergo full developmental and biochemical acclimation over seasons. The integration of this mechanistic model with canopy-level photosynthesis and crop growth models could greatly enhance our ability to predict crop performance under variable climate scenarios.

## 5. Conclusions

This study provides quantitative evidence that sweet sorghum actively modulates its photosynthetic light-use strategy under conditions of elevated CO_2_ and temperature. Through mechanistic modeling, we demonstrated that elevated CO_2_ enhances effective light absorption capacity (*σ′*_ik_) and electron transport rate (*J*_max_), while high temperature activates photoprotective mechanisms such as NPQ, accompanied by a reduction in excited-state pigment pool size (*N*_k_). These coordinated adjustments stabilize PSII efficiency (Φ_PSII_) under high irradiance, thereby minimizing photoinhibition and strengthening photosynthetic resilience. The tight coupling between Φ_PSII_ and *σ′*_ik_ underscores a fine-tuned regulatory mechanism that balances light harvesting with energy dissipation. Our findings also highlight the utility of mechanistic models for deciphering biophysical responses of C_4_ crops to combined abiotic stress. The identified parameters—*σ*′_ik_, *N*_k_, *τ*_min_—offer potential biomarkers for screening and selecting genotypes with enhanced photosynthetic plasticity. Overall, sweet sorghum exhibits a remarkable capacity to optimize its photosynthetic apparatus under climate-relevant stressors, highlighting its potential contribution to future food and bioenergy security. Future research should validate these mechanisms under field conditions and assess their long-term impacts on biomass production and sugar yield. Integrating these mechanistic traits into breeding programs and crop modeling frameworks will be essential for developing climate-resilient C_4_ crops.

## Figures and Tables

**Figure 1 biology-14-01185-f001:**
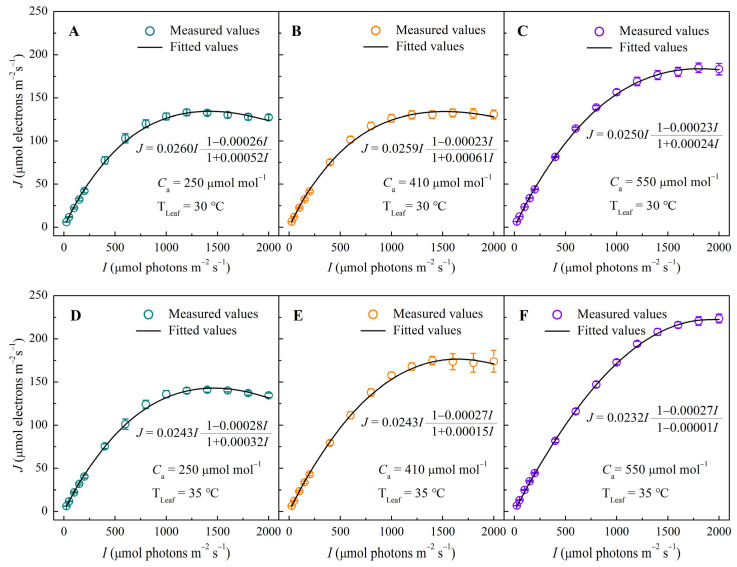
Response curves of electron transport rate (*J*) to irradiance (*I*) in sweet sorghum under CO_2_ concentrations of 250, 410, and 550 μmol mol^−1^ at 30 °C (panels **A**–**C**) and at 35 °C (panels **D**–**F**).

**Figure 2 biology-14-01185-f002:**
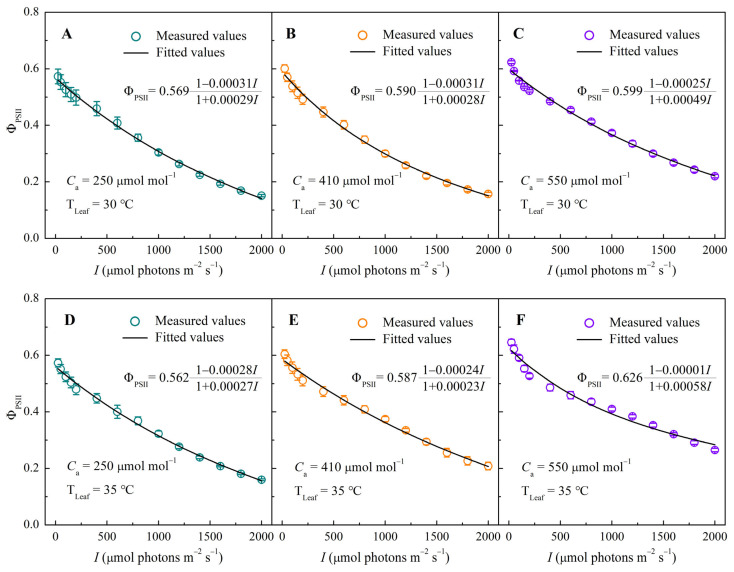
Response curves of effective quantum efficiency (Φ_PSII_) to irradiance (*I*) in sweet sorghum under CO_2_ concentrations of 250, 410, and 550 μmol mol^−1^ at 30 °C (panels **A**–**C**) and at 35° C (panels **D**–**F**).

**Figure 3 biology-14-01185-f003:**
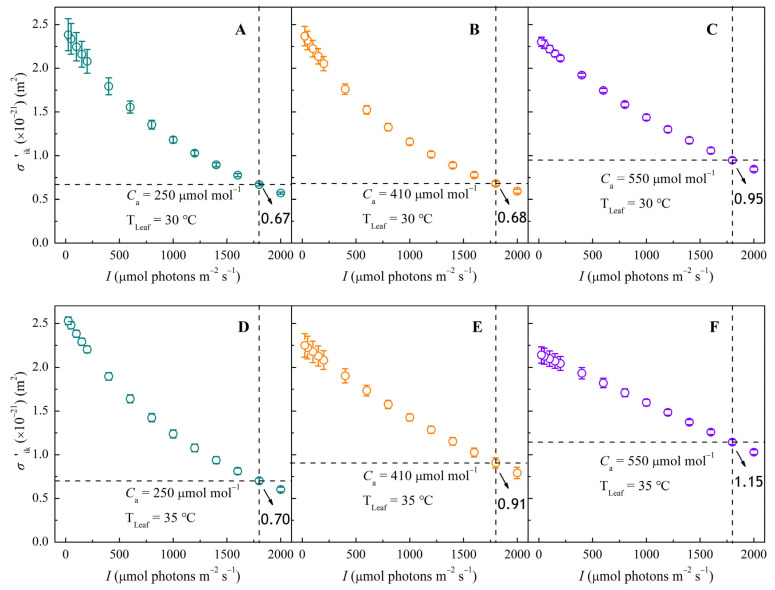
Response curves of effective absorption cross-section of pigment molecules (*σ′*_ik_) to irradiance (*I*) in sweet sorghum under CO_2_ concentrations of 250, 410, and 550 μmol mol^−1^ at 30 °C (panels **A**–**C**) and at 35 °C (panels **D**–**F**).

**Figure 4 biology-14-01185-f004:**
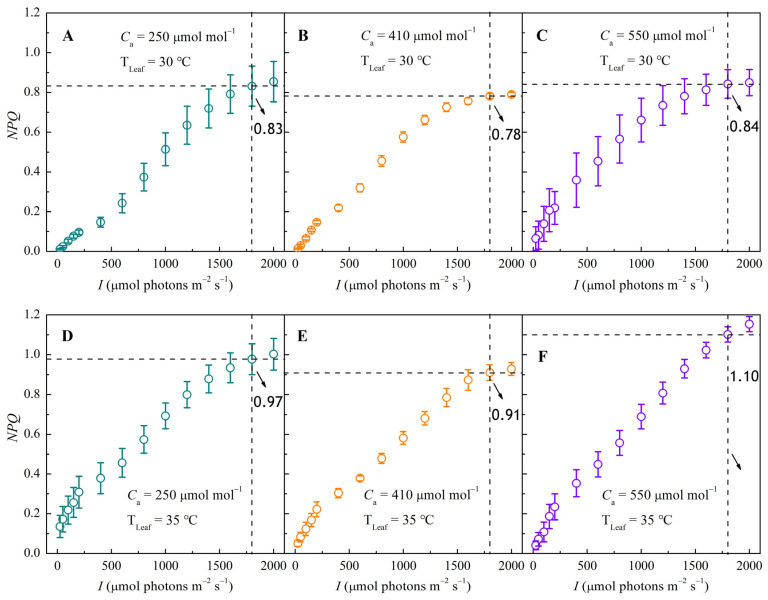
Response curves of non-photochemical quenching (NPQ) to irradiance (*I*) in sweet sorghum under CO_2_ concentrations of 250, 410, and 550 μmol mol^−1^ at 30 °C (panels **A**–**C**) and at 35 °C (panels **D**–**F**).

**Figure 5 biology-14-01185-f005:**
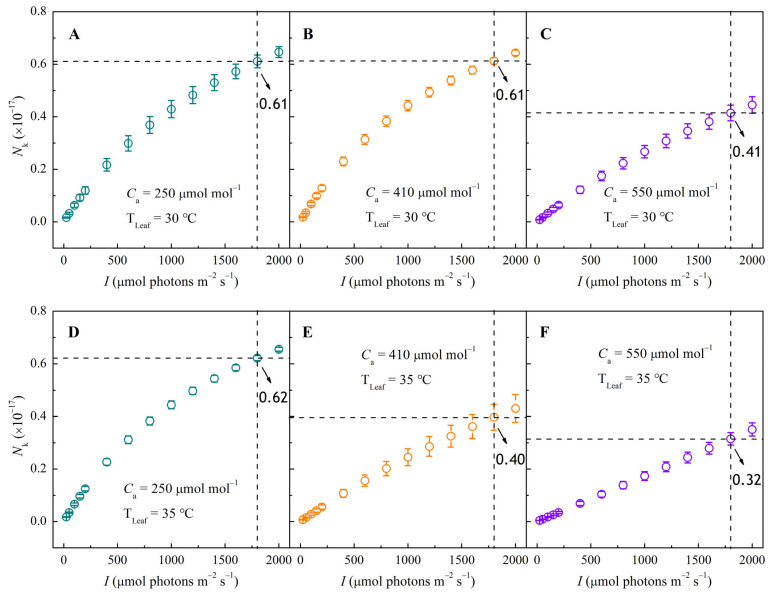
Response curves of excited-state pigment molecules (*N*_k_) to irradiance (*I*) in sweet sorghum under CO_2_ concentrations of 250, 410, and 550 μmol mol^−1^ at 30 °C (panels **A**–**C**) and at 35 °C (panels **D**–**F**).

**Table 1 biology-14-01185-t001:** Photosynthetic parameters derived from *J*–*I* curves and Φ_PSII_–*I* response curves of sweet sorghum under varying ambient CO_2_ concentration (*C*_a_) and leaf temperature (T_Leaf_) conditions.

Photosynthetic Parameters	T_Leaf_ = 30 °C						T_Leaf_ = 35 °C					
	*C*_a_ = 250 μmol mol^−1^		*C*_a_ = 410 μmol mol^−1^		*C*_a_ = 550 μmol mol^−1^		*C*_a_ = 250 μmol mol^−1^		*C*_a_ = 410 μmol mol^−1^		*C*_a_ = 550 μmol mol^−1^	
	Fitted	Measured	Fitted	Measured	Fitted	Measured	Fitted	Measured	Fitted	Measured	Fitted	Measured
*α*	0.262 ± 0.020 ^ab^	―	0.261 ± 0.013 ^ab^	―	0.252 ± 0.005 ^ab^	―	0.278 ± 0.004 ^a^	―	0.246 ± 0.016 ^ab^	―	0.232 ± 0.009 ^b^	―
*J*_max_ (μmol electron m^−2^ s^−1^)	135.46 ± 3.53 ^c^	133.63 ± 3.30 ^c^	134.67 ± 5.05 ^c^	133.84 ± 5.52 ^c^	184.27 ± 5.93 ^b^	184.82 ± 5.49 ^b^	141.48 ± 6.46 ^c^	142.20 ± 1.92 ^c^	179.55 ± 8.72 ^b^	179.88 ± 8.45 ^b^	223.16 ± 4.90 ^a^	223.57 ± 5.06 ^a^
*I*_e-sat_ (μmol photons m^−2^ s^−1^)	1412.1 ± 43.4 ^d^	1280.0 ± 48.9 ^d^	1515.0 ± 33.3 ^cd^	1600.0 ± 63.2 ^c^	1795.7 ± 16.5 ^b^	1800.0 ± 0.0 ^b^	1411.4 ± 17.6 ^d^	1400.0 ± 63.2 ^d^	1665.4 ± 154.8 ^bc^	1666.7 ± 176.4 ^bc^	1896.5 ± 28.9 ^ab^	2000.00 ± 0.00 ^a^
*σ*_ik_ (×10^−21^ m^2^)	2.43 ± 0.19 ^ab^	―	2.42 ± 0.12 ^ab^	―	2.33 ± 0.06 ^ab^	―	2.58 ± 0.05 ^a^	―	2.28 ± 0.14 ^b^	―	2.16 ± 0.09 ^b^	―
*τ*_min_ (ms)	7.83 ± 0.71 ^a^	―	8.65 ± 0.61 ^a^	―	5.14 ± 0.52 ^b^	―	7.88 ± 0.62 ^a^	―	4.68 ± 0.44 ^b^	―	2.93 ± 0.23 ^c^	―
*N*_0_ (×10^16^ m^2^)	17.09 ± 0.06 ^a^	―	17.10 ± 0.07 ^a^	―	13.99 ± 0.09 ^b^	―	17.09 ± 0.06 ^a^	―	13.29 ± 0.12 ^b^	―	13.51 ± 0.96 ^b^	―
Φ_PSIImax_	0.569 ± 0.029 ^b^	0.573 ± 0.026 ^b^	0.590 ± 0.018 ^a^	0.601 ± 0.013 ^a^	0.599 ± 0.003 ^b^	0.623 ± 0.005 ^ab^	0.562 ± 0.018 ^b^	0.573 ± 0.015 ^b^	0.587 ± 0.018 ^b^	0.604 ± 0.015 ^ab^	0.626 ± 0.009 ^a^	0.646 ± 0.011 ^a^
*R*^2^ (*J*–*I* curves)	0.9970 ± 0.0006	―	0.9979 ± 0.0005	―	0.9994 ± 0.0001	―	0.9972 ± 0.0007	―	0.9975 ± 0.0018	―	0.9996 ± 0.0001	―
*R*^2^ (*Φ*_PSII_–*I* curves)	0.9945 ± 0.0014	―	0.9932 ± 0.0023	―	0.9889 ± 0.0034	―	0.9895 ± 0.0039	―	0.9900 ± 0.0031	―	0.9770 ± 0.0009	―

Parameters include the maximum electron transport rate (*J*_max_), saturation irradiance (*I*_e-sat_), total number of pigment molecules in the measured leaf (*N*_0_), eigen-absorption cross-section of pigment molecules (*σ*_ik_), minimum average lifetime of pigment molecules in the excited state (*τ*_min_), and the maximum effective quantum efficiency of PSII (Φ_PSIImax_). All values are presented as means ± *SE*. Different letters indicate statistically significant differences (*p* < 0.05) between fitted and measured values within each treatment.

## Data Availability

The original contributions presented in this study are included in the article. Further inquiries can be directed to the corresponding authors.

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
