# Peer review of "Regulation of Light Absorption and Energy Dissipation in Sweet Sorghum Under Climate-Relevant CO2 and Temperature Conditions"

_biology, 2025, doi:10.3390/biology14091185_

Round 1
Reviewer 1 Report
Comments and Suggestions for Authors
Regulation of light absorption and energy dissipation in sweet sorghum under elevated CO2 and temperature
In the current manuscript, the authors have tried to give us a comprehensive picture about how sweet sorghum orchestrates molecular-scale adaptations to maintain photosynthetic function under varying CO2 concentrations (250, 410, and 550 μmol mol−1) and temperatures (30 °C and 35 °C). Additionally, the obtained results can give us an expected picture to how sweet sorghum copes with different future climate scenarios specifically which are related to the efficiency of photosynthesis and enhancing plant productivity. Several aspects have been studied either as measured traits or fitted by a simulation. In my opinion the overall quality of this manuscript is good and written well; it is written in logical sequence and used latest references. However, I think it needs minor revision before considering this manuscript satisfactory for publishing.
# Some identified parameters should be presented in the abstract section as percentages.
# In the M&M section (line 146), the authors estimated the concentration of chlorophyll. However, no results were presented in the results section.
# Some studied relationships as shown in Figure 1 did not exactly follow the linear model. So i suggest the authors should provide the equation on each figure (Fig1-5) to understand the mathematical nature of each equation beside its R2
# Achieving the redox balance in sweet sorghum plants under high temperatures and elevating the concentration of CO2 need to be further elucidated in depth either in the results section or in the discussion.
# Line 336 PEPcase instead of PEPC
Author Response
Reviewer 1 Comments: In the current manuscript, the authors have tried to give us a comprehensive picture about how sweet sorghum orchestrates molecular-scale adaptations to maintain photosynthetic function under varying CO2 concentrations (250, 410, and 550 μmol mol−1) and temperatures (30 °C and 35 °C). Additionally, the obtained results can give us an expected picture to how sweet sorghum copes with different future climate scenarios specifically which are related to the efficiency of photosynthesis and enhancing plant productivity. Several aspects have been studied either as measured traits or fitted by a simulation. In my opinion the overall quality of this manuscript is good and written well; it is written in logical sequence and used latest references. However, I think it needs minor revision before considering this manuscript satisfactory for publishing.
Response: We deeply appreciate your highly professional and constructive feedback. These thoughtful critiques and suggestions have been invaluable for substantially improving and strengthening our work. We have carefully revised our manuscript to address each point raised and hope the changes made meet with approval. The revisions made to the manuscript are distinctly highlighted through the utilization of blue-colored text in our revised submission. A point-by-point elucidation of our responses to your comments is presented as follows.
Comment 1: Some identified parameters should be presented in the abstract section as percentages.
Response 1: We thank the reviewer for the constructive suggestion. We have revised the abstract to include quantitative percentage changes for key parameters (Jmax, σ′ik, NPQ, and Nk), providing a clearer and more impactful summary of our findings. The revised abstract is as follows:
“Understanding how environmental factors regulate photosynthetic energy partitioning is crucial for enhancing crop resilience in future climates. This study investigated the light-response dynamics of sweet sorghum (Sorghum bicolor L. Moench) leaves under combinations of CO2 concentrations (250, 410, and 550 μmol mol−1) and temperatures (30 °C and 35 °C), using integrated chlorophyll fluorescence measurements and mechanistic photosynthesis modeling. Our results revealed that elevating CO2 from 250 to 550 μmol mol−1 significantly increased the maximum electron transport rate (Jmax) by up to 57%, and enhanced the effective light absorption cross-section (σ′ik) by 64% under high light and elevated temperature (35 °C), indicating improved photochemical efficiency and light-harvesting capability. Concurrently, these adjustments reduced PSII down-regulation. Increased temperature stimulated thermal dissipation, reflected in a rise in non-photochemical quenching (NPQ) by 0.13–0.26 units, accompanied by a reduction in the number of excited-state pigment molecules (Nk) by 20-33%. The strongly coordinated responses between quantum yield (ΦPSII) and σ′ik highlights a dynamic balance among photochemistry, heat dissipation, and fluorescence. These findings elucidate the synergistic photoprotective and energy-partitioning strategies that sweet sorghum employs under combined CO2 enrichment and heat stress, providing mechanistic insights for optimizing photosynthetic performance in C4 crops in a changing climate.”
Comment 2: In the M&M section (line 146), the authors estimated the concentration of chlorophyll. However, no results were presented in the results section.
Response 2: We thank you for this insightful comment. We apologize for this omission. We have added the description of chlorophyll content in the second paragraph of the 3.1 results section and analyzed it in combination with the intrinsic characteristics of chlorophyll molecules. Added content is also given as follows: “Integration of chlorophyll content data (425.90 ± 1.67 mg m−2) into the J–I model revealed a decline in the eigen-absorption cross-section (σik) with rising CO2, especially at 35°C (from 2.58 ± 0.05 × 10−21 m2 to 2.16 ± 0.09 × 10−21 m2).” (To see lines 263-265 on page 9, marked in blue)
Comment 3: Some studied relationships as shown in Figure 1 did not exactly follow the linear model. So I suggest the authors should provide the equation on each figure (Fig1-5) to understand the mathematical nature of each equation beside its R2.
Response 3: We appreciate this excellent suggestion. We have now updated Figures 1 and 2 to include the specific best-fit equations. However, Figures 3 to 5 do not have fitted curves. NPQ is the actual value measured by the instrument, and σ′ik and Nk are parameter values calculated through the J-I relationship combined with chlorophyll content; the parameters in their relationship are the same as those in the J-I relationship. Therefore, we have not updated Figures 3 to 5.
Comment 4: Achieving the redox balance in sweet sorghum plants under high temperatures and elevating the concentration of CO2 need to be further elucidated in depth either in the results section or in the discussion.
Response 4: We sincerely thank the reviewer for this insightful comment, which has helped us significantly strengthen the mechanistic interpretation of our findings. As suggested, we have enhanced these discussions in section 4.1, "Regulation of Electron Transport and Light Utilization," as follows:
“Dynamic down-regulation of PSII is an adaptive reduction in photochemical efficiency triggered by supra-saturating irradiance, mediated largely through energy-dependent quenching (qE) and sustained NPQ [29, 30]. This mechanism protects the electron transport chain by dissipating excess energy and maintaining redox balance [31]. In our study, photoinhibition and PSII down-regulation were pronounced under low CO2, but were strongly suppressed under elevated CO2. Although high temperature typically disrupts the electron transport chain and induces redox imbalance, particularly under combined high light and heat stress [18, 32]. Our results indicate that elevated CO2 partially alleviate this inhibition at 35 °C. Specifically, the upward shift in J–I curve and sustained J under high CO2 suggest improved plastoquinone (PQ) turnover and redox balance, delaying photoinhibition. The significant increase in Jmax under elevated CO2, even at 35°C, reflects enhanced consumption of ATP and NADPH by the Calvin cycle. This increased metabolic demand likely prevents over-reduction of electron carriers such as PQ pool and ferredoxin, by providing an efficient sink for photosynthetic products. In contrast, elevated temperature alone intensified excitation pressure, as evidenced by increased NPQ and decreased Nk in the present study. However, the synergistic effect between high CO2 and temperature appeared to mitigate this effect through stimulated carbon assimilation, which consumed more reductants and helped maintain linear electron flow. Thus, sweet sorghum appears to exploit elevated CO2 to stabilize electron transport and counteract heat-induced stress, reducing reliance on photoprotective energy dissipation.” (To see lines 351–370 on Page 14, marked in blue)
Comment 5: Line 336 PEPcase instead of PEPC.
Response 5: We thank the reviewer for pointing this out. The abbreviation has been corrected to "PEPcase" as suggested on line 336 of the revised manuscript.
Reviewer 2 Report
Comments and Suggestions for Authors
Sweet sorghum is a representative C4 crop, which has a strong adaptability to high temperatures and elevated CO2 levels. The photoprotective and energy-use strategies of sweet sorghum under combined CO2 and heat stress were analyzed in this paper to predict the adaptability of C4 crop under the future greenhouse condition. This research is urgently needed and has innovative value in agriculture. But the language and design of this paper need to be improved. The research content and level of the paper are suitable for publication in Biology Journal after revision. Some suggestions were put forward as follows.
- Titile: It is suggested to delete "temperature" in the title. The treatment temperatures in this research are 30 ℃ and 35 ℃, which are normal temperatures in summer and not high-temperature stress. For C4 plants, the high-temperature stress should be at least 40 ℃ or above.
- Abstract: “we quantified key physiological parameters including electron transport rate (J), effective quantum yield of PSII (PSII), non-photochemical quenching (NPQ), effective light absorption cross-section (σ′ik), and the number of excited-state pigment molecules (Nk).” The sentence above is unnecessary. The logical relationship among various photosynthetic parameters in the results section of the abstract is not clear enough.
“Results revealed that” should revise to “Our results revealed that”
- Keyword part: Delete "temperature" in the Keyword selection. “photosynthetic pigment molecules” should be revise to “photosynthetic pigments”
- Introduction part: “while the accompanying……” in line 50. It is suggested to modify to “together with the ATP synthesis driving by proton gradient.
Language expression of this paper requires polishing by native language experts.
- Materials and Methods part:
(1) In line 124. The unit of light intensity is Lux, and the unit of light intensity in line 133 is μmol m−2 s−1. The unit of light intensity should be uniform
(2) Please explain the basis and reasons for choosing the CO2 concentration of 250, 410, and 550 μmol mol−1.
(3) Please explain the basis and reasons for choosing 30 ℃ and 35℃. Only when the temperature is around 45 ℃ can cause stress to sorghum. The results show that photosynthetic activity is higher at 35 ℃ than that of at 30 ℃ in Figure 1.
(4) the flow rate of 500 μmol s–1 in Line 142-143, Is this flow rate the flow rate of the mixed gas or that of CO2?
(5) “One-way analysis of variance (ANOVA)” in Line 209 should be “Two-way analysis of variance” . Temperature and CO2 concentration
- Results part:
- The title of Table 1 is placed above the table, and the table notes are placed below it.Set the page to landscape and place the data on the same row.
- The figures are not clear enough.
- Footnote part:
It is suggested that all the abbreviated symbols and their physiological meanings be listed in the footnote.
Comments on the Quality of English LanguageSweet sorghum is a representative C4 crop, which has a strong adaptability to high temperatures and elevated CO2 levels. The photoprotective and energy-use strategies of sweet sorghum under combined CO2 and heat stress were analyzed in this paper to predict the adaptability of C4 crop under the future greenhouse condition. This research is urgently needed and has innovative value in agriculture. But the language and design of this paper need to be improved. The research content and level of the paper are suitable for publication in Biology Journal after revision. Some suggestions were put forward as follows.
- Titile: It is suggested to delete "temperature" in the title. The treatment temperatures in this research are 30 ℃ and 35 ℃, which are normal temperatures in summer and not high-temperature stress. For C4 plants, the high-temperature stress should be at least 40 ℃ or above.
- Abstract: “we quantified key physiological parameters including electron transport rate (J), effective quantum yield of PSII (PSII), non-photochemical quenching (NPQ), effective light absorption cross-section (σ′ik), and the number of excited-state pigment molecules (Nk).” The sentence above is unnecessary. The logical relationship among various photosynthetic parameters in the results section of the abstract is not clear enough.
“Results revealed that” should revise to “Our results revealed that”
- Keyword part: Delete "temperature" in the Keyword selection. “photosynthetic pigment molecules” should be revise to “photosynthetic pigments”
- Introduction part: “while the accompanying……” in line 50. It is suggested to modify to “together with the ATP synthesis driving by proton gradient.
Language expression of this paper requires polishing by native language experts.
- Materials and Methods part:
(1) In line 124. The unit of light intensity is Lux, and the unit of light intensity in line 133 is μmol m−2 s−1. The unit of light intensity should be uniform
(2) Please explain the basis and reasons for choosing the CO2 concentration of 250, 410, and 550 μmol mol−1.
(3) Please explain the basis and reasons for choosing 30 ℃ and 35℃. Only when the temperature is around 45 ℃ can cause stress to sorghum. The results show that photosynthetic activity is higher at 35 ℃ than that of at 30 ℃ in Figure 1.
(4) the flow rate of 500 μmol s–1 in Line 142-143, Is this flow rate the flow rate of the mixed gas or that of CO2?
(5) “One-way analysis of variance (ANOVA)” in Line 209 should be “Two-way analysis of variance” . Temperature and CO2 concentration
- Results part:
- The title of Table 1 is placed above the table, and the table notes are placed below it.Set the page to landscape and place the data on the same row.
- The figures are not clear enough.
- Footnote part:
It is suggested that all the abbreviated symbols and their physiological meanings be listed in the footnote.
Author Response
Reviewer 2 Comments: Sweet sorghum is a representative C4 crop, which has a strong adaptability to high temperatures and elevated CO2 levels. The photoprotective and energy-use strategies of sweet sorghum under combined CO2 and heat stress were analyzed in this paper to predict the adaptability of C4 crop under the future greenhouse condition. This research is urgently needed and has innovative value in agriculture. But the language and design of this paper need to be improved. The research content and level of the paper are suitable for publication in Biology Journal after revision. Some suggestions were put forward as follows.
Response: Thank you for your constructive feedback on our manuscript. We appreciate your recognition of the study’s strengths and have carefully addressed your suggestions to enhance clarity and focus. Revisions are highlighted in blue text. Our point-by-point responses to your comments are detailed below.
Titile: It is suggested to delete "temperature" in the title. The treatment temperatures in this research are 30 ℃ and 35 ℃, which are normal temperatures in summer and not high-temperature stress. For C4 plants, the high-temperature stress should be at least 40 ℃ or above.
Response: We appreciate your suggestion. While extreme heat can occur in some sweet sorghum growing regions of China, its duration is short. Previous studies indicate that temperatures exceeding the optimum (around 30 °C for sweet sorghum) constitute heat stress. Therefore, we chose 35 °C to simulate a moderately elevated, ecologically relevant temperature, reflecting anticipated higher summer averages under climate change. To avoid misunderstanding, we have revised the title to "Regulation of light absorption and energy dissipation in sweet sorghum under climate-relevant CO2 and temperature conditions".
Abstract: “we quantified key physiological parameters including electron transport rate (J), effective quantum yield of PSII (PSII), non-photochemical quenching (NPQ), effective light absorption cross-section (σ′ik), and the number of excited-state pigment molecules (Nk).” The sentence above is unnecessary. The logical relationship among various photosynthetic parameters in the results section of the abstract is not clear enough. ” Results revealed that” should revise to “Our results revealed that”
Response: We appreciate your suggestion. As suggested, we have modified the Abstract for clarity, as follows:
“Understanding how environmental factors regulate photosynthetic energy partitioning is crucial for enhancing crop resilience in future climates. This study investigated the light-response dynamics of sweet sorghum (Sorghum bicolor L. Moench) leaves under combinations of CO2 concentrations (250, 410, and 550 μmol mol−1) and temperatures (30 °C and 35 °C), using integrated chlorophyll fluorescence measurements and mechanistic photosynthesis modeling. Our results revealed that elevating CO2 from 250 to 550 μmol mol−1 significantly increased the maximum electron transport rate (Jmax) by up to 57%, and enhanced the effective light absorption cross-section (σ′ik) by 64% under high light and elevated temperature (35 °C), indicating improved photochemical efficiency and light-harvesting capability. Concurrently, these adjustments reduced PSII down-regulation. Increased temperature stimulated thermal dissipation, reflected in a rise in non-photochemical quenching (NPQ) by 0.13–0.26 units, accompanied by a reduction in the number of excited-state pigment molecules (Nk) by 20-33%. The strongly coordinated responses between quantum yield (ΦPSII) and σ′ik highlights a dynamic balance among photochemistry, heat dissipation, and fluorescence. These findings elucidate the synergistic photoprotective and energy-partitioning strategies that sweet sorghum employs under combined CO2 enrichment and heat stress, providing mechanistic insights for optimizing photosynthetic performance in C4 crops in a changing climate.”
Keyword part: Delete "temperature" in the Keyword selection. “photosynthetic pigment molecules” should be revise to “photosynthetic pigments”.
Response: Thank you for pointing this out. We have removed “temperature” from the keyword list and revised “photosynthetic pigment molecules” to “photosynthetic pigments.”
Introduction part: “while the accompanying……” in line 50. It is suggested to modify to “together with the ATP synthesis driving by proton gradient. Language expression of this paper requires polishing by native language experts.
Response: We agree with the reviewer’s recommendation. The expression has been modified to “together with the ATP synthesis driven by proton gradient”. The manuscript has also undergone professional language editing.
Materials and Methods part:
(1) In line 124. The unit of light intensity is Lux, and the unit of light intensity in line 133 is μmol m−2 s−1. The unit of light intensity should be uniform
Response: We appreciate your careful reading. The unit of light intensity has been unified throughout the manuscript as μmol photons m−2 s−1.
(2) Please explain the basis and reasons for choosing the CO2 concentration of 250, 410, and 550 μmol mol−1.
Response: We appreciate the reviewer’s comment. Before the Industrial Revolution, the global CO2 concentration was approximately 270 μmol mol−1, and it has gradually increased to about 415 μmol mol−1 at present. According to the IPCC reports, the concentration of CO2 is continuously rising at an annual rate of about 2–3 μmol mol−1, and it is projected to reach around 550 μmol mol−1 by the mid-21st century. Therefore, the selected levels of 250, 410, and 550 μmol mol−1 in our study represent pre-industrial, current, and future projected CO2 concentrations, respectively.
Reference: IPCC. Climate Change 2021 – The Physical Science Basis: Working Group I Contribution to the Sixth Assessment Report of the Intergovernmental Panel on Climate Change, Cambridge: Cambridge University Press, 2023.
(3) Please explain the basis and reasons for choosing 30 °C and 35 °C. Only when the temperature is around 45 °C can cause stress to sorghum. The results show that photosynthetic activity is higher at 35 °C than that of at 30 °C in Figure 1.
Response: Thank you for this valuable comment. Previous studies have suggested that when the ambient temperature exceeds the optimum temperature for plant growth, it can be regarded as a form of heat stress (Liu et al., 2002). For sweet sorghum, the optimal temperature is about 30 °C. Although extreme heat (>40 °C) does occur in some regions of China where sweet sorghum is cultivated (e.g., Northeast, Inner Mongolia, Gansu, Xinjiang), the duration of such extreme events is relatively short. Thus, we chose 35 °C to represent a moderately elevated temperature above the optimum, which is ecologically relevant and reflects the higher average summer temperatures expected under climate change scenarios.
Reference: Liu, D.H., Zhao, S.W., Gao, R.F., et al. Response of plant photosynthesis to high temperature. Bull Botan Res, 2002, 22(2): 205–212 (in Chinese with English abstract)
(4) the flow rate of 500 μmol s–1 in Line 142-143, Is this flow rate the flow rate of the mixed gas or that of CO2?
Response: We thank the reviewer for pointing this out. The reported flow rate (500 μmol s–1) refers to the total flow rate of the mixed gas (air + CO2). CO2 was supplied from an external gas cylinder and mixed with air by the LI-6800 system. We have revised the manuscript accordingly to clarify this point (To see lines 159-160 on Page 4, marked blue).
(5) “One-way analysis of variance (ANOVA)” in Line 209 should be “Two-way analysis of variance”. Temperature and CO2 concentration
Response: Thank you for noting this. The statistical method has been corrected to “Two-way ANOVA” (To see lines 224-228 on Page 6, marked blue).
Results part: The title of Table 1 is placed above the table, and the table notes are placed below it. Set the page to landscape and place the data on the same row. The figures are not clear enough.
Response: We agree with your suggestion. The title of Table 1 has been placed above the table, and the notes are placed below. The page has been adjusted to landscape format and the numbers have been enlarged.
Reviewer 3 Report
Comments and Suggestions for Authors
This study investigated the ability of sweet sorghum to dynamically regulate its photosynthetic response to elevated and depressed CO2 and temperature. A strength of the study is that the authors applied mechanistic models to evaluate the biophysical responses of sorghum to abiotic stress. Overall, the manuscript is well structured, the materials and methods are described in sufficient detail, and the conclusions are consistent with the results.
There are several issues:
1. Photosynthetic responses to different light intensities have been previously studied in sweet sorghum under different temperatures and CO2 conditions http://dx.doi.org/10.3389/fpls.2024.1291630. The design of this study is very similar to that presented in this manuscript. The same analytical models and calculations were used. For example, the earlier study measured the net photosynthetic rate, while this manuscript used the electron transport rate, which correlates well with the net photosynthetic rate. The authors did not indicate that this is a continuation or addition to the previous study. The authors should emphasize the novelty of this study; if this is a continuation of an earlier work, this should be indicated. The discussion section should compare the results with the earlier study.
2. In my opinion, the title of the manuscript does not correspond to the design of the study. This study studies not only the effect of elevated CO2 and temperature, but also the effect of reduced CO2 levels.
3. The authors should point out the limitations of this study. The plants used for the experiments were not adapted to high CO2 and high temperature, so the effects may be short-term. Of course, the LI-6800 portable photosynthesis system is a good tool for research, but adaptation of plants to stress conditions before measurements also contributes to the adequacy of the results. Field studies with long-term adaptation of plants to stress conditions should confirm the results. In this study, the results will be valid only for short-term stress.
Author Response
Reviewer 3 Comments: This study investigated the ability of sweet sorghum to dynamically regulate its photosynthetic response to elevated and depressed CO2 and temperature. A strength of the study is that the authors applied mechanistic models to evaluate the biophysical responses of sorghum to abiotic stress. Overall, the manuscript is well structured, the materials and methods are described in sufficient detail, and the conclusions are consistent with the results.
Response: We greatly appreciate Reviewer 3’s careful reading of our manuscript and the positive feedback regarding the structure, methodological detail, and conclusions. We are particularly grateful for the recognition of our use of mechanistic modeling to investigate photosynthetic regulation. We will continue to refine the presentation of our findings to ensure clarity and scientific rigor in future work.
Question 1: Photosynthetic responses to different light intensities have been previously studied in sweet sorghum under different temperatures and CO2 conditions http://dx.doi.org/10.3389/fpls.2024.1291630. The design of this study is very similar to that presented in this manuscript. The same analytical models and calculations were used. For example, the earlier study measured the net photosynthetic rate, while this manuscript used the electron transport rate, which correlates well with the net photosynthetic rate. The authors did not indicate that this is a continuation or addition to the previous study. The authors should emphasize the novelty of this study; if this is a continuation of an earlier work, this should be indicated. The discussion section should compare the results with the earlier study.
Response: We thank the reviewer for raising this important point. While both studies used the same plant material, growth conditions, and experimental setup to ensure comparability, they investigate fundamentally different physiological processes and mechanistic questions.
Our previous work (Yang et al., 2024, Front. Plant Sci.) focused on gas exchange parameters (e.g., Aâ‚™, gâ‚›, Táµ£, WUE) and compared empirical vs. mechanistic models for simulating light-response curves. The primary outcome was model validation and water-use efficiency dynamics.
The present study employs chlorophyll fluorescence techniques and a mechanistic model of photosynthetic electron transport to investigate molecular-scale processes such as electron transport rate (J), effective quantum yield of PSII (ΦPSII), non-photochemical quenching (NPQ), effective absorption cross-section (σ′ᵢₖ), number of excited-state pigment molecules (Nâ‚–). These parameters provide deeper insights into photoprotective mechanisms and energy partitioning strategies at the pigment-molecule level, which were not addressed in our previous work.
We have now explicitly highlighted these differences in the Introduction and Discussion sections to emphasize the novel contributions of this study. In introduction, we have added “Building upon our previous research which characterized the leaf-scale gas exchange responses (net photosynthetic rate An, stomatal conductance gS, transpiration rate Tr, and water-use efficiency WUE) of sweet sorghum to varying light, CO2, and temperature [13], this study delves deeper into the underlying photobiological mechanisms. Our earlier work established how these environmental factors affect the overall carbon and water flux but did not address the molecular-scale processes governing light energy absorption, transfer, and dissipation.” (To see lines 96-102 on Page 3, marked blue). In Discussion section, we have also modified to “This study extends our previous work on sweet sorghum, which characterized leaf-scale gas exchange responses (e.g., An, gs, and WUE) to light, CO2, and temperature [13]. Here, by integrating chlorophyll fluorescence with a mechanistic biophysical model, we elucidated the molecular-scale mechanisms underlying those leaf-level responses. Our findings reveal how sweet sorghum coordinates CO2 availability, temperature, and irradiance to sustain photosynthetic function, thereby providing a mechanistic explanation for previously observed physiological patterns.” (To see lines 331-337 on Page 13, marked blue).
Question 2: In my opinion, the title of the manuscript does not correspond to the design of the study. This study studies not only the effect of elevated CO2 and temperature, but also the effect of reduced CO2 levels.
Response: Thank you for your suggestion. Before the Industrial Revolution, the global CO2 concentration was approximately 270 μmol mol−1, then it has gradually increased to about 415 μmol mol−1 at present. According to the IPCC reports, the concentration of CO2 is continuously rising at an annual rate of about 2–3 μmol mol−1, and it is projected to reach around 550 μmol mol−1 by the mid-21st century. Therefore, the selected levels of 250, 410, and 550 μmol mol−1 in our study represent pre-industrial, current, and future projected CO2 concentrations, respectively. For temperature, previous studies have suggested that when the ambient temperature exceeds the optimum temperature for plant growth, it can be regarded as a form of heat stress (Liu et al., 2002). The optimal temperature of sweet sorghum is about 30 °C. Although extreme heat (>40 °C) does occur in some regions of China where sweet sorghum is cultivated (e.g., Northeast, Inner Mongolia, Gansu, Xinjiang), the duration of such extreme events is relatively short. Thus, we chose 35 °C to represent a moderately elevated temperature above the optimum, which is ecologically relevant and reflects the higher average summer temperatures expected under climate change scenarios.
We have changed the title to accurately reflect the full range of COâ‚‚ and temperature conditions tested. Revised Title is “Regulation of light absorption and energy dissipation in sweet sorghum under climate-relevant CO2 and temperature conditions”.
IPCC. Climate Change 2021 – The Physical Science Basis: Working Group I Contribution to the Sixth Assessment Report of the Intergovernmental Panel on Climate Change, Cambridge: Cambridge University Press, 2023.
Liu, D.H., Zhao, S.W., Gao, R.F., et al. Response of plant photosynthesis to high temperature. Bull Botan Res, 2002, 22(2): 205–212 (in Chinese with English abstract)
Question 3: The authors should point out the limitations of this study. The plants used for the experiments were not adapted to high CO2 and high temperature, so the effects may be short-term. Of course, the LI-6800 portable photosynthesis system is a good tool for research, but adaptation of plants to stress conditions before measurements also contributes to the adequacy of the results. Field studies with long-term adaptation of plants to stress conditions should confirm the results. In this study, the results will be valid only for short-term stress.
Response: We appreciate the reviewer's insight. We have clarified in the Discussion's final paragraph that our study captures acclimatory physiological responses, not long-term adaptive changes. The specific additions are: “However, it is important to note that the responses reported in this study reflect short-term acclimation to sudden changes in CO2 and temperature, as the plants were not pre-adapted to these stress conditions. While these immediate mechanistic responses are highly informative for understanding the initial biophysical adjustments of photosynthesis, they may differ from long-term adaptations that involve changes in gene expression, leaf morphology, and nutrient allocation. For instance, long-term exposure to elevated CO2 can lead to photosynthetic acclimation via carbohydrate accumulation and nitrogen dilution [14]. Therefore, field studies employing Free-Air CO2 Enrichment or open-top chambers are essential to validate these mechanisms under realistic conditions where plants undergo full developmental and biochemical acclimation over seasons.” (To see lines 421-430 on Page 15, marked blue)
Round 2
Reviewer 3 Report
Comments and Suggestions for Authors
The authors took into account all the comments and supplemented the manuscript. I have no more questions.